# Responses of Spring Discharge to Different Rainfall Events for Single-Conduit Karst Aquifers in Western Hunan Province, China

**DOI:** 10.3390/ijerph18115775

**Published:** 2021-05-27

**Authors:** Wei Chang, Junwei Wan, Jiahua Tan, Zongxing Wang, Cong Jiang, Kun Huang

**Affiliations:** 1School of Environmental Studies, China University of Geosciences, 388 Lumo Road, Wuhan 430074, China; changwei@cug.edu.cn (W.C.); wanjw@cug.edu.cn (J.W.); wangzongxing@cug.edu.cn (Z.W.); jiangcong@cug.edu.cn (C.J.); 2China Railway Siyuan Survey and Design Group Co., Ltd., 745 Peace Avenue, Wuhan 430063, China; tsytanjiahua@126.com

**Keywords:** karst spring, flow recession curve, precipitation intensity, recession coefficient

## Abstract

It is a challenge to describe the hydrogeological characteristics of karst aquifers due to the complex structure with extremely high heterogeneity. As the response of karst aquifers to rainfall events, spring discharge variations after precipitation can be used to identify the internal structure of karst systems. In this study, responses of spring discharge to different kinds of precipitations are investigated by continuously monitoring precipitation and karst spring flow at a single-conduit karst aquifer in western Hunan province, China. Recession curves were used to analyze hydrodynamic behaviors and separate recession stages. The results show that the shape of the recession curve was changed under different rainfall conditions. Recession processes can be divided in to three recession stages under heavy rain conditions due to water drainage mainly from conduits, fracture, and matrix at each stage, but only one recession stage representing drainage mainly from matrix in the case of light rain. With the change in amount and intensity of precipitation, the calculated recession coefficient at each stage changes in an order of magnitude. The influence of precipitation on the recharge coefficient and the discharge composition at each recession are discussed, and then the conceptual model diagram of water filling and releasing in the single-conduit karst aquifers is concluded. The findings provide more insight understand on hydraulic behaviors of karst spring under different types of rainfall events and provide support for water resource management in karst regions.

## 1. Introduction

Karst water is a precious freshwater resource that feeds about one quarter of the world’s population and will also play a strategic role in economic and social relationships in the future [1,2]. Fast flow to the groundwater through focused recharge is known to transmit short-lived pollutants into carbonate aquifers, endangering the quality of groundwaters where one quarter of the world’s population lives [3]. It is still a challenge to predict the distribution and quantity of water resources in karst aquifers due to their complex structures with extremely high heterogeneity and dramatic variability in groundwater dynamic [4,5]. Carbonate formation that has undergone karstification can form a network of various scale gaps, such as caves, conduits, fractures, and pores, which can be described as dual-porosity or triple-porosity aquifers [6,7]. Conduit systems with high permeability, but limited volume, mainly act as preferential pathways for transferring groundwater. However, the matrix systems (fractures and pores) with relatively lower permeability but more interspace act as reservoirs for storing groundwater [8,9]. The physical processes of flow in karst aquifers are primarily determined by the characteristic of the complex conduit-matrix systems, therefore, the responses of spring discharge to precipitation can be used to study the hydrodynamic functioning of karst systems and identify aquifer structures and hydraulic parameters.

Many methods are applied to identify karst groundwater systems, such as hydrogeochemical analysis [10], environmental isotope method [11,12,13], time-series analysis [14,15] and artificial tracer tests [16,17]. Tracer tests provide the most effective means for identifying the point-to-point connections between flow inputs (sinkholes or sinking streams) and outputs (springs).Tracer test can not only identify the flow path of karst groundwater, but also can calculate the geometric parameters (transport velocities and dispersion) of groundwater flow, which is helpful to analyze the internal structure of karst aquifers [18]. The volume of an aquifer can be estimated by analyzing the breakthrough curve of a tracer test and the change in spring discharge, and the diameter of the conduit can also be estimated on the assumption that water flows in a cylindrical conduit [19].

Spring hydrography provides information on karst aquifer recharge and responses to rainfalls, which shows a rapid increase limb and then a slower decrease limb. Spring recession curves are a vital tool for identifying hydrodynamic behaviors during a period without precipitation and estimating permeability by analyzing the shape of the curves [2,20,21,22,23,24]. Boussinesq and Maillet proposed different models based on different physical models to analyze the hydrological recession process; the Boussinesq model is a quadratic equation, and the Maillet model is a simple exponential equation. Both of the two models are used to describe the whole recession process of the discharge from an aquifer, but different flow regimes can hardly be noticed [24,25]. Therefore, a series of improved equations are proposed to describe the decay process of karst springs. A modified Maillet model was always used; it can be expressed by a sum of several exponential components, and each exponential component represents a type of medium release water, such as conduits, fractures, and matrix [24,26,27]. Unlike the modified Maillet equation, Mangin divides the composition of karst spring into two parts: fast flow and slow flow. The exponential equation was suitable for modeling the baseflow, but not for the quick flow [24,28]. However, distinguishing the flow of different media was difficult using the modified Maillet model, and distinguishing the fast flow and slow flow using the Mangin model was difficult. A model of dividing the discharge recession process into several stages according to the flow regimes is also proposed, each recession stage is subject to exponential attenuation law, and different recession stages represent the conduit, fracture or matrix medium control of the release of water [18,22,29]. In hydrological analysis, it is customary to represent the recession curves graphically with a semi-logarithmic plot, the discharge (Q) is plotted on the logarithmic ordinate and time (t). A complete log(Q)-t chart of hydrological recession process may have multiple straight lines, and generally there are three straight line stages [20,30]. In the first stage, spring discharge is considered mainly from the conduit reservoir through quick flow and, in the third stage, spring discharge is mainly from the matrix called baseflow [22,23,24,26,31]. The middle stage is defined as the fracture-drainage stage, with spring discharge from the fracture, or as mixture-drainage stage, with spring discharge from both conduit and matrix [22,23,26]. Flow exchange between different reservoirs is determined by the hydraulic head difference [32,33,34].

Spring recession curves analysis were widely used to study the hydrodynamic behaviors and hydraulic property of karst aquifers, providing insight for spring discharge responses to precipitation events. Nevertheless, the spring recession processes are greatly influenced by the precipitation pattern, even for a particular karst system, which is still in need of further research to reveal spring discharge responses to different types of rainfalls. For instance, during heavy rain, point-concentrated recharge through sinkholes rapidly fills into the conduits and causes a high level of hydraulic head in the conduit at the peak of spring hydrograph [35]; but, during small rainfall amounts, area-averaged infiltration is dominant through fractures, resulting in a high level of hydraulic head in matrix but low level in conduits at the peak discharge. Therefore, the recession processes at the spring show significant difference under various rainfalls, due to the water distribution condition at the beginning of the recession processes. It may provide more information for recognizing hydraulic property variation in the vertical direction and estimating the effective porosity of karst aquifers by analyzing spring recession curves together with thermal responses under various rainfall events [6,24,26].

In this paper, the responses of spring discharge to rainfall events were investigated by the continuous monitoring of spring discharge and temperature at a single-conduit karst system. The recharge area and conduit characteristics were verified through tracer tests during several precipitation events. The spring recession processes were analyzed after a series of precipitation events, and the reasons for the change in water recession regular patterns are discussed under different rainfall conditions. Finally, conceptual models are proposed to explain the hydraulic property and responses of the karst flow system to rainfall events. As the drinking water source of the local downstream residents, the research results can provide a guide for water resource management and water quality protection at Daiye cave or other similar karst spring systems.

## 2. Study Area

The study area (28°47′08.4″–28°48′17.8″ N, 110°00′59.4″–11°02′12.8″ E) is located in the Yongshun county of northwest Hunan province, China (Figure 1), and is in a typical subtropical monsoon climate with four distinct seasons.

The average annual rainfall is 1400 mm a year, with about 60% of precipitation concentrated in the rainy season from April to August. The geological structure of the study area is uniclinal, mainly consisting of Middle and Upper Cambrian that tilts to the northwest with the stratigraphic dip at about 30 degrees. The lithology of the Upper Cambrian is mainly limestone and dolomite with strong karstification, developing various karst depressions and sinkholes in the surface and, correspondingly, karst conduits and subsurface rivers underground. The lithology of the Middle Cambrian is mainly noncarbonate or argillaceous carbonate with weak permeability, which acts as a waterproof floor for the Upper Cambrian karst aquifer systems.

Daiye cave and Lanhua cave are two concentrated discharge points of the Upper Cambrian karst aquifers (Figure 1a); a previous study [36] identified the groundwater cycle characteristics of these two caves. Lanhua cave is a complex underground river system with multiple conduits, which can be divided into two parts, divided by the skylight LHD (Q_1_). In the downstream of Q_1_, the conduit scale is quite large and already verified by cave measurements (Figure 1b); however, in the upstream of Q_1_, the conduit scale is relatively small and filled with water. According to hydrogeological survey and previous tracer tests, the sinking streams 1, 2, and 3 flow into the Lanhua cave system and sinking stream 4 flows into the Daiye cave system [36]. The tracer tests confirmed that the Daiye cave groundwater system recharged by the karst platform with an elevation of 600–1000 m above sea level, about 3.74 km^2^ recharge area [36]. A creek, originated from a series of epikarst springs, dives into the karst conduit through a sinkhole at the altitude about 700 m (Figure 1), and then discharges to the surface at Daiye cave, with an elevation of 660 m and with an average hydraulic gradient around 3.39% for the underground river. Therefore, the Daiye cave system included both karst conduit with extremely high permeability, which transports quick flow, and matrix, including fissures and pores with relatively lower permeability, that transports slow flow. Especially during intense fall periods, the creek not only gathers the upstream springs, but also converges a large amount of slope flow around the depression, which results in a rapid increase in spring discharge at Daiye cave.

## 3. Materials and Methods

### 3.1. Monitoring of Precipitation and Spring Discharge

To explore the aquifer structure characteristics of the Daiye Cave karst groundwater system, precipitation and spring discharge were continually monitored from July 2016 to July 2018. Rainfalls were observed by an RG3-M rain gauge (HOBO Onset, Bourne, MA, USA), with an accuracy of 0.2 mm. Spring discharge at Daiye cave was indirectly obtained by monitoring the water level variations every 20 min at the artificial weir, using a pressure sensor (Model 3001 LTC Levelogger, Solinst Canada Ltd., Georgetown, ON, Canada). During the monitoring period, the maximum daily rainfall was 137 mm/d in the study area, and the spring discharge at Daiye cave ranged from 10 to 2100 L/s.

### 3.2. Theoretical

#### 3.2.1. Spring Recession Analysis

The method of characterizing karst springs is based on the exponential equation (1). Discharge recession curves were divided into several stages based on flow regimes [18]: (1)$$Q_{t}=Q_{0}e^{-\alpha (t-t_{0})}$$
where *t* is any time since the beginning of the recession for which discharge is calculated, α is the recession coefficient, *t_0_* is the time at the beginning of the recession (usually set equal to zero), *Q_t_* is the spring discharge at time *t*, *Q*_0_ is spring discharge at the start of the recession (*t*_0_).

Generally speaking, the recession curve can be divided into three sections, which represent different media and mainly control the water release process: quick flow (conduit-dominated flow), slow flow (diffuse-dominated flow), or mixed flow [18,22,26]. Equation (1) can be rewritten as Equation (2).
(2)$$Q_{t}=\left\{ \begin{array}{l} Q_{1}e^{-\alpha_{1}t}(0\leq t\leq t_{1})\\ Q_{2}e^{-\alpha_{2}t}(t_{1}\leq t\leq t_{2})\\ Q_{3}e^{-\alpha_{3}t}(0\leq t\leq t_{3}) \end{array}\right.$$

In the first recession stage (conduit-dominated flow stage), the discharge includes water released from conduits, fractures, and matrix media. In the second recession stage (mixed flow stage), the discharge includes water release from fractures and matrix media. In the third stage (diffuse-dominated flow), the discharge is entirely released by matrix medium. The change in water release of each medium with time is shown in Equation (3).
(3)$$\left\{ \begin{array}{l} Q_{c}=Q_{1}e^{-\alpha_{1}t}-Q_{2}e^{-\alpha_{2}t}(0\leq t\leq t_{1})\\ Q_{f}=Q_{2}e^{-\alpha_{2}t}-Q_{3}e^{-\alpha_{3}t}(0\leq t\leq t_{2})\\ Q_{m}=Q_{3}e^{-\alpha_{3}t}(0\leq t\leq t_{3}) \end{array}\right.$$

The relationship between the discharge and time of each medium is calculated by Equation (4).
(4)$$k_{i}=100\times Q_{i}/(Q_{c}+Q_{f}+Q_{m})$$

Water storage capacity during different recession stages can be calculated from this model, using the Equation (5) [29]: (5)$$\left\{ \begin{array}{l} V_{c}=\int_{0}^{t_{1}}\left(Q_{1}e^{-\alpha_{1}t}-Q_{2}e^{-\alpha_{2}t}\right)dt\\ V_{f}=\int_{0}^{t_{2}}\left(Q_{2}e^{-\alpha_{2}t}-Q_{3}e^{-\alpha_{3}t}\right)dt\\ V_{m}=\int_{0}^{t_{3}}\left(Q_{3}e^{-\alpha_{3}t}\right)dt \end{array}\right.$$
where *V_i_* is the volume drained during period *t_i_*.

Recession coefficient is a comprehensive reflection of hydraulic conductivity and water storage capacity, which can be calculated by Equation (6) [22]:
(6)$$\alpha =\pi^{2}T/4SL^{2}$$
where *T* is transmissivity, *S* is storativity, *L* is distance from the discharge point to the drainage divide.

#### 3.2.2. Tracer Tests

To investigate the groundwater flow velocity and the geometric parameters, two groups of tracer tests were carried out in the Daiye cave system. Uranine was chosen as tracer and injected into the Xiazhai sinkhole after two rainfall events, with different flow rates in the conduit during the period of 18–24 May 2018 (Table 1 and Figure 1a). The concentrations of uranine were measured by a fluorometer (GGUN-FL Fluorometer, Neuchâtel Switzerland) at the Daiye cave spring every 10 min with the accuracy around 0.01 ppb.

160 g and 370 g sodium fluorescein were injected into the same sinkhole at different flow rates. The first group of tracer tests was carried out under the condition of an average discharge of 60 L/s without rainfall before tracer injection, while the second group of artificial tracer test was carried out with an average discharge of 650 L/s after a 36.8 mm rainfall.

The observed tracer concentration breakthrough curve can be used to calculate the tracer recovery ratio *R* as follows [19] (Equation (7)):(7)$$R=\int_{0}^{\infty }C\left(t\right)Q\left(t\right)dt/M_{0}$$
where *M*_0_ is the mass of the injected tracer, *C*(*t*) is the tracer concentration at time *t* during the test, *Q*(*t*) is spring discharge of Daiye cave at time *t*. According to the breakthrough curve, one can also obtain the average flow velocity ($\bar{v}$) in the conduit by Equation (8) as follows:(8)$$\bar{v}=x_{s}/t_{0}$$
in which *x_s_* is the distance between the injection and recovery points, *t*_0_ is the mean transit time. Discharge was measured during each tracer test, allowing for a rough estimate of aquifer volume by the tracer cloud using Equation (9), and the cross-sectional area can be estimated from Equation (10).
(9)$$V=\int_{0}^{\bar{t}}Qdt$$
(10)$$A=V/x_{s}$$
where *V* is conduit volume, *A* is the mean cross-sectional area, and $\bar{t}$ is the duration of tracer tests.

Assuming the karst conduit to be a cylindrical channel, the flow-channel diameter (*D_c_*) can be estimated from Equation (11), where *D_c_* represents the conduit diameter in the Daiye cave system.
(11)$$D_{C}=2\sqrt{A/\pi }$$

## 4. Results

### 4.1. Breakthrough Curves for Tracer Tests under Different Precipitation Conditions

Figure 2 shows the variations in tracer concentration with time during the two group of tracer tests. The two breakthrough curves of tracer tests are both single peak curves, in accordance with the single conduit structure of the Daiye cave system. The average travel velocity of tracer test in the conduit is 251.65 and 32.72 m/h, indicating that the conduit runoff is very rapid and unstable.

Groundwater flow rate significantly affects the shape of the tracer breakthrough curve. For tracer test 1, the duration time is longer and the concentration peak is smaller than tracer test 2 (Figure 2) due to a longer retention time and more sufficient diffusion in the conduit at lower flow rate, which results in a longer tail during the recession process of tracer concentration [37] and a lower recovery rate.

The straight-line distance from the underground river entrance to exit (tracer migration) is about 1180 m. Conduit geometric parameters were calculated based on tracer breakthrough curves using Equation (7) to Equation (9), with similar results for both tests (Table 2). The calculated average diameter of karst conduit for Daiye cave system is about 2.687 m, and the total volume of the conduit is 6692 m^3^.

### 4.2. Spring Discharge Variations at Daiye Cave

Figure 3 plots the time series of precipitation, spring discharge, and groundwater temperature for the period from May 2016 to May 2018. The range of discharge is 10–2100 L/s, and the range of the groundwater temperature is 9.7–18.9 °C. Spring discharge and groundwater temperature quickly respond to rainfalls only a few hours later.

10 rainfall–discharge response curves were selected, including 6 heavy rains (P ≥ 25 mm/d) and 4 light rains (0 ≤ P < 25 mm/d); the principle of rainfall–discharge response curve selection was that rainfall is relatively concentrated, and there is no rainfall event 5 days after the occurrence of the rainfall event, as shown in Figure 4. Spring discharge rises rapidly within a few hours after rain, and the groundwater temperature also changed abruptly due to rainfall recharge. Under the conditions of heavy rain (H1–H6), discharge of the Daiye cave spring has a steep rise and fall; however, for light rain (L7–L10), discharge curves are much wider and gentler with multiple lower peaks. Rainfall–discharge curves also indicate that the Daiye cave conduit is unobstructed; therefore, rainfall recharge will quickly produce a discharge response.

Only one peak appears in the rainfall–discharge curves of H1–H5; among which, rainfall events in H1 and H5 have a pause process where the discharge first rises to a certain amount and then stabilizes for a short time. The reason for the pause phenomenon is that the distribution of rainfall mainly concentrates in two periods with a rain pause period that is too short for the discharge turning into the recession process for the former rainfall; meanwhile, a stronger discharge response to the subsequent rainfall already occurs, resulting in a new rising period of discharge at Daiye cave. Nevertheless, for rainfalls in H2 and H3, representative single peak curves were obtained with a sharp rise stage followed by a relatively gentle descent stage, corresponding to more concentrated rainfall events. Besides, the discharge curve for rainfall H6 shows three incremental peaks during the ascent stage (Figure 4), due to three concentrated and strong rainfall periods during the rainfall event.

Table 3 shows the characteristic parameters of rainfall–discharge response curves under 10 rainfall events, where lag time represents the time from the moment of maximum rainfall to peak discharge and delay time represents the time from the discharge response to the end of the flow decline. In case of heavy rain (H1–H6), discharge responds within 2 h after the rainfall event, and reaches the peak flow within 5 h, with a lag time of 1–5 h and delay time of 71–161 h. Under light rain conditions (L7–L10), the response times are about 4–9 h, and peak flows appear 17–28 h later, with a lag time of 18–29 h and a delay time of 110–144 h.

### 4.3. Recession Processes under Different Rainfall Conditions

Table 4 shows the fitting results of 10 groups of discharge recessions. Discharge recession curves can be decomposed into three exponential stages in the case of heavy rain, such as H2 (P = 41.8 mm) shown in Figure 5. With the decrease in rainfall, two exponential stages can fit the recession curve. However, only one exponential stage can fit the recession curve in the case of light rain, such as L8 (P = 18.2 mm) shown in Figure 5.

According to the recession curves, the volume of water released from each medium was calculated, shown in Table 5. For heavy rainfalls (H1–H6), the volume of water from the conduit accounts for 5.2–15.1% with a duration of 4–10 h, and water from the fracture accounts for 8.0–20.8% with a duration of 14.7–27.7 h. Except for H4, the volume of water discharged from fracture accounts for more than 70% of the total discharge. However, for light rainfalls (L7–L10), spring discharge is released entirely from the matrix.

## 5. Discussion

### 5.1. Influence of Precipitation on Discharge Composition of Recession Process

As mentioned above, the discharge recession curves for heavy rain (H1–H6) can be divided into two or three stages (Table 4). In the first stage of the recession process, spring discharge is mainly composed of water released from conduit that moves as quick flow and generally reaches the outlet several hours after rainfalls, with the recession coefficient ranging from 0.121 to 0.331 (Table 5). Then, spring discharge is mainly composed of water released from fracture after the conduit water drainage, which is the middle stage, with the coefficient ranging from 0.03 to 0.102 (Table 5). In the third stage, water release from matrix medium is dominant, which feed to fractures and then enter the conduit, and the conduits act as the water conduction channel. As water flow in matrix is relatively slow and the total volume is quite large compared with conduit and fracture water, the recession stage of matrix water is much longer, always lasting over 100 h, and smoother, with a relatively more stable recession coefficient ranging from 0.009 to 0.022 (Table 5).

To investigate the variation in water composition during the recession processes, the proportion of water released from conduit, fracture, and matrix under heavy rains (H1–H6) were calculated by Equation (4) and plotted in Figure 6. It shows that the ratio of water release from conduit decreases rapidly, while the ratio of water release from matrix gradually increases and becomes predominant. The variation in the ratio of water release from fracture is more complicated, which generally rises first and then declines after peak, except for rainfall event H4 where the ratio of fracture water continually decreases. According to the discharge recession analysis, the water release from conduit is 2436–11,655 m^3^, which exceeds the total volume of conduit (6691.6 m^3^) calculated according to tracer tests for rainfall event H5 and H6. It indicates that the conduit may be fully filled with water and pressure flow may last for a period due to the continual surface runoff from the upstream of Xiazhai sinkhole under the long period (over 30 h) of heavy rain (over 60 mm).

The variation in recession behaviors may be induced by the double recharge manner of both planar infiltration through fractures and point injection through sinkholes, and the regulation and storage of multiple karst aquifer media (conduits, fractures, and matrix) under different patterns of precipitation. Under light rain conditions, planar infiltration was the main recharge method when the rainfall intension does not exceed the infiltration capacity. Infiltration flow moves slowly in the small pores and fractures that mainly play the roles of storage space, and then converges into large fractures and conduits that primarily play the roles of transmissivity channels. Therefore, the discharge curves (L7-L10) are fairly smooth, with a low peak and long tail during the recession process (Figure 4) that is controlled by water release from matrix medium (Table 5). However, point injection recharge plays the primary role when the rainfall intensity exceeds the infiltration capacity under heavy rain conditions. The surface runoff formed in the upstream of the Xiazhai sinkhole rapidly injects into conduit and causes a steep rise in discharge at the outlet of Daiye cave (Figure 4).

Water temperature variations with discharge also provide corresponding evidence. Figure 7 shows typical water temperature variations under heavy rain conditions in the wet season (summer), when the temperature of precipitation is higher than that of groundwater. Groundwater temperature quickly forms a rising pulse signal after rainfall due to the fast recharge of higher temperature precipitation through concentrated injection into conduit at the Xiazhai sinkhole and shows a fluctuation decline in the discharge recession process until it returns to the groundwater temperature before the rainfall event (Figure 7). Similarly, Figure 8 reveals the temperature changes in groundwater that are mainly influenced by planar infiltration recharge under light rain in the dry season (winter) when the temperature of groundwater is higher than precipitation. Groundwater temperature exhibits a slight downtrend as the discharge increases due to the slow and low temperature water flow recharged by precipitation infiltration (Figure 8).

### 5.2. Influence of Precipitation on Infiltration Coefficient and Recession Coefficient

Rainfall pattern has a significant impact on the recharge manner and infiltration coefficients (Figure 9). The values of calculated infiltration coefficient gradually decrease as the rainfall amount and intensity increase. Then, the infiltration coefficient maintains a relative stabilization during rainfall amounts larger than 40 mm (Figure 9a) or rainfall intensities larger than 4 mm/h (Figure 9b). However, the situation for heavy rain (H5) is an exception, for which the rainfall amount exceeds 60 mm but the rainfall intensity is about 2 mm/h and may be suitable for infiltration at Daiye cave system. When rainfall intensity is larger than 2 mm/h, the proportion of surface runoff that flows to the outside of the system increases and results in a lower infiltration coefficient.

The recession coefficient is a comprehensive reflection of hydraulic conductivity and water storage capacity, which is also related to the water-filling state of the karst aquifer system at the initial point of recession curves [22]. As the peak water levels in in conduits and matrix are variational under different rainfall events, it may influence the values of the calculated recession coefficient even for similar kinds of rainfalls. Figure 10 plots a recession coefficient with rainfall amount and rainfall intensity, showing different relationships for each recession phase. Only one recession stage appears for light rain, L7–L10, (P ≤ 25 mm) with the rainfall intensity less than 2 mm/h, and the values of recession coefficient α_3_ vary between 0.01 and 0.1, showing an increasing trend with rainfall amount. Two or three recession stages were found for heavy rain, H1–H4, (40 ≤ P ≤ 45 mm) and torrential rain, H5 and H6, (P ≥ 50 mm). Generally, the values of the recession coefficients α_1_ and α_2_ decrease as the rainfall amount increases; however, the values of recession coefficients show a positive relationship with rainfall intensity under similar rainfall amounts, such as heavy rain, H1-H4. It is worth noting that the variations of α_3_ are more complicated, where the calculated values of α_3_ for light rain are obviously higher than that of heavy rain and torrential rain. Meanwhile, the values of α_3_ for torrential rain, H5 and H6, are greater than that of heavy rain, H1–H4.

To further investigate the influence of the water-filling state of the karst aquifer system on recession coefficients, the relationship between the initial discharge flow rate (*Q_t_*) and the recession coefficient (α) at different stages of the Daiye cave spring were plotted in Figure 11, including observations under 10 independent rainfalls (flow without superposition) and other non-independent rainfalls (flow with superposition). It shows that the data points of short-term heavy rainfall are mainly near curve I, where the values of recession coefficient increase rapidly with the increase in discharge. Correspondingly, the data points of long-term heavy rainfall are near curve II, where the values of recession coefficient increase more smoothly with the increase in discharge compared with curve I. It indicates that the values of recession coefficients for short-term heavy rain are greater than that for long-term heavy rain even though the initial discharge flow is the same, which may be explained by the differences in the water filling state under various precipitations. For short-term heavy rainfall, concentrate injection recharge is the dominant recharge and replenishment of matrix is not insufficient; thus, water release is controlled by conduit and a large scale of fractures. For long-term heavy rainfall, both concentrate injection and planar infiltration are important and the replenishment of the karst aquifer system is more sufficient for both conduit and matrix; hence, the discharge recession process is controlled by conduit, fracture, and matrix in sequence. Therefore, curve Ⅰ and curve Ⅱ represent, respectively, an insufficient replenishment system, where water filling in conduits is predominant, and a sufficient replenishment system, with water filling in conduit, fracture, and matrix. Other data points are in the middle of curve Ⅰ and curve Ⅱ, where water filling states are moderate.

In summary, the recharge and discharge characteristics of the Daiye aquifer system under different rainfall conditions are illustrated by Figure 12, Figure 13 and Figure 14. For ease of discussion, rainfalls were divided into three types: ①, short-time heavy rainfall (large rainfall intensity, small total rainfall, such as H1-H4); ②, long-term heavy rainfall (medium rainfall intensity, large total rainfall, such as H5 and H6); and ③, light rain (P < 25 mm).

In the case of short-time heavy rainfall, characterized by a strong rainfall intensity but a limited rainfall amount (such as H1-H3), the water level of conduit increases rapidly after precipitation and higher than that of fractures and matrix (Figure 12b), as the concentrated injection at the sinkhole predominates [32,38]. Therefore, the water discharged mainly comes from conduit medium at the early stage, with a relatively large value of recession coefficients α_1_ and α_2_ (Figure 12b). However, the values of the recession coefficient (α_3_) are relatively small in the third stage (Figure 12c), due to the water filling in fracture and the matrix medium lagging and being insufficient.

For long-term heavy rainfall with a medium rainfall intensity and large rainfall amount (such as H6), both conduit and matrix can obtain sufficient recharge through centralized injection and planar infiltration (Figure 13b). The water levels of conduit, fracture, and matrix all stay at a high position at the beginning of recession process, after an adequate water exchange between different kinds of aquifer media. Compared with heavy rainfall (H1-H3), water levels of conduit and fracture may be lower at the peak flow point, resulting in smaller values of recession coefficient α_1_ and α_2_. Nevertheless, water levels of matrix are higher during the third recession stage on account of the quick discharge rate of conduit and fracture water (Figure 11c). Correspondingly, the values of α_3_ are greater for long-term heavy rainfall (H5 and H6) than that of heavy rainfall.

Under the condition of light rain with small rainfall intensity and rainfall amount (such as L7–L10), the aquifer system is mainly recharged by planar infiltration. Therefore, only fracture and matrix obtain effective supply, but the conduit mainly plays a role of a drainage gallery with low water level (Figure 14b). Spring discharge recession processes are mainly controlled by the fracture and matrix medium, appearing at only one recession stage with the values of the recession coefficient between α_2_ and α_3_ for heavy rainfall.

### 5.3. Limitation of This Study

The limitation of this study is that the rain gauge station is not in the recharge area of the Daiye cave system but is instead located 2.8 km away from the southeast direction of the sinkhole. The variation in microclimate in mountainous areas may have a certain influence on the results. Besides, the conceptual mode of water filling and release processes in the Daiye cave system was proposed based on an analysis of spring discharge monitoring data, which still exits in an uncertain way and needs to be further verified by borehole water levels.

## 6. Conclusions

In this study, the response of spring discharge to rainfall events was investigated by continuously monitoring precipitation and karst spring flow at Daiye cave, a representative single-conduit karst system in western Hunan province, China. The distribution of the karst conduit was verified by tracer tests, with the average diameter of 2.687 m estimated by analyzing the tracer concentration breakthrough curve. Recession curves were used to analyze hydrodynamic behaviors and separate recession stages. The results show that the shape of the recession curve was changed under different rainfall conditions. Recession processes can be divided into three recession stages under heavy rain conditions due to water drainage mainly from conduits, fracture, and matrix at each stage with the ratio of 5.2–15.1% conduit water and 8–20.8% fracture water, but only one recession stage representing drainage mainly from matrix in the case of light rain. An interesting finding is that the calculated recession coefficient at each stage is not a constant and changes in an order of magnitude but is related to different amounts and intensities of precipitation. Recession coefficients decrease with increasing durations of precipitation events for the same amount of rainfall, which could be attributed to various response times to precipitation for different kinds of aquifer media. The water filling and releasing speed of conduit medium is obviously faster than that of fracture and matrix medium; therefore, the rainfall intensity controls the speed of water filling of medium and the rainfall duration controls the saturation degree of the aquifers. When precipitation intensity exceeds infiltration capacity, surface runoff concentrates into karst aquifers through the sinkholes, resulting in a quick response in discharge at the spring outlet and steep slope in the recession curves due to fast flow in conduits. On the contrary, when precipitation intensity is lower than infiltration capacity, the aquifer is mainly recharged through planar infiltration in fractures, causing a higher response time to precipitation at the spring due to the slow flow in the fractures and matrix. Finally, a typical recharge and discharge model diagram of karst water system in Southwest China is put forward for different rainfall conditions. Under light rain, it is mainly surface dispersed infiltration recharge, while, in heavy rain conditions, it is mainly concentrated recharge. These findings can provide more insight on hydraulic behaviors of karst springs under different types of rainfall events and scientific support for water resource management and utilization.

## Figures and Tables

**Figure 1 ijerph-18-05775-f001:**
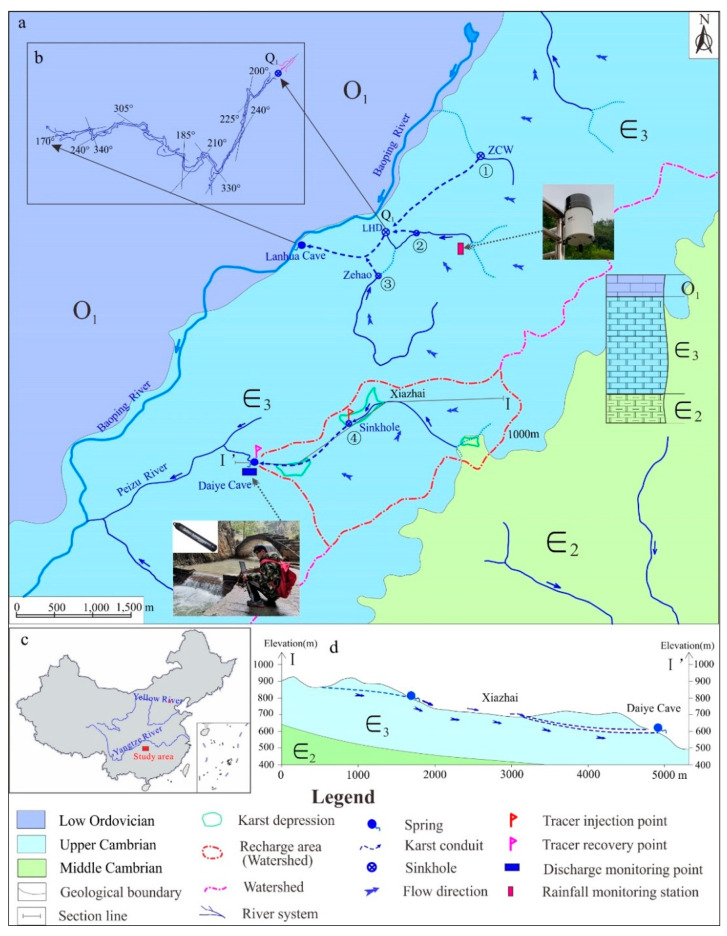
(**a**) Hydrogeological map of study area; (**b**) Measured underground river direction map; (**c**) Location of the study site in China; (**d**) Schematic geological-hydrogeological cross section of the I–I’ line shown in (**a**).

**Figure 2 ijerph-18-05775-f002:**
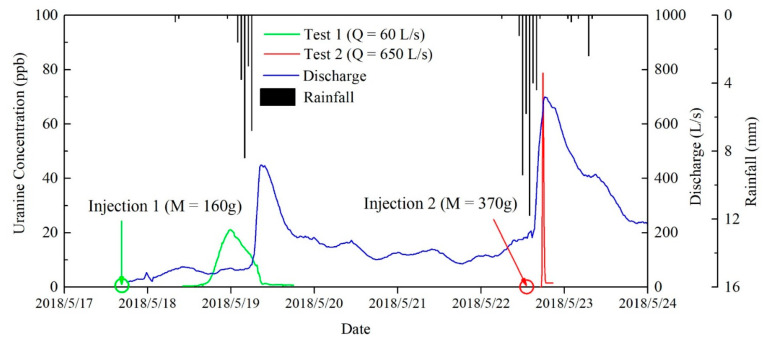
Breakthrough curves of tracer tests.

**Figure 3 ijerph-18-05775-f003:**
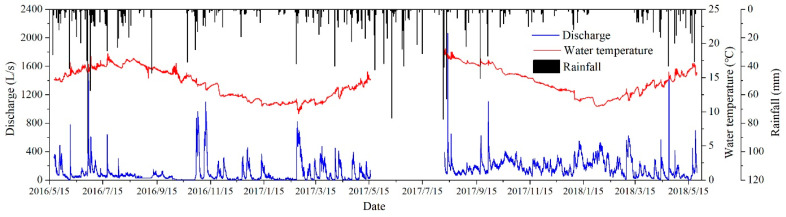
Precipitation–discharge–water-temperature response of Daiye cave system.

**Figure 4 ijerph-18-05775-f004:**
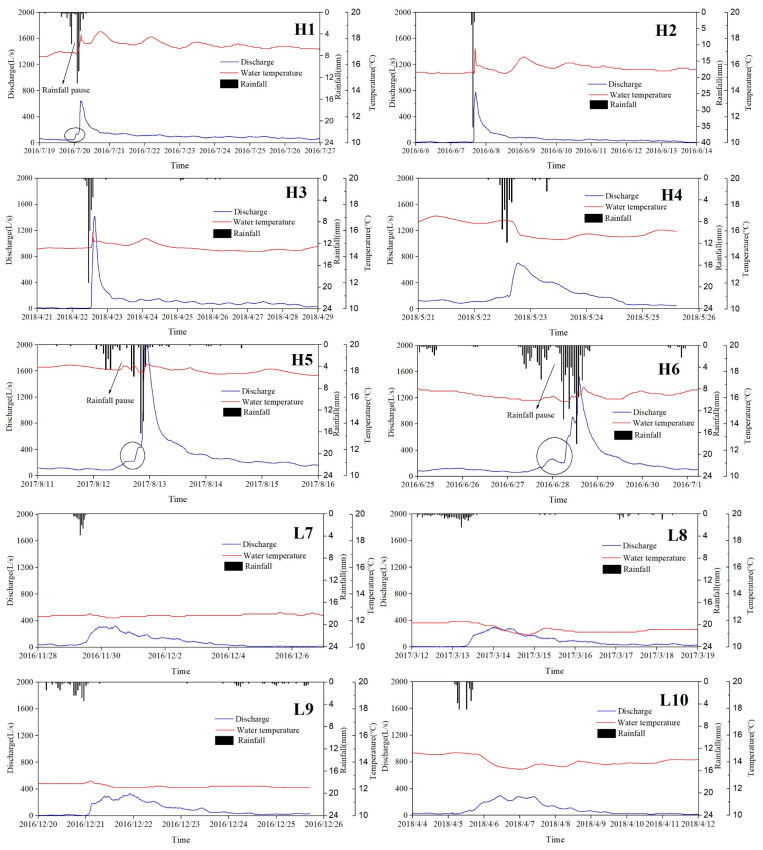
Rainfall–discharge–water-temperature response curve of 10 precipitations.

**Figure 5 ijerph-18-05775-f005:**
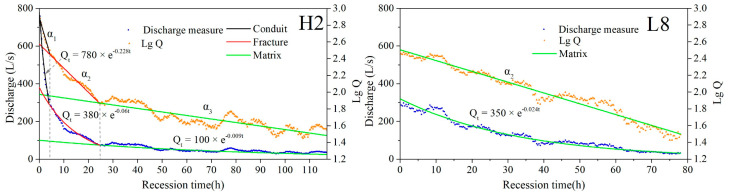
Recession analysis under two different rainfall patterns.

**Figure 6 ijerph-18-05775-f006:**
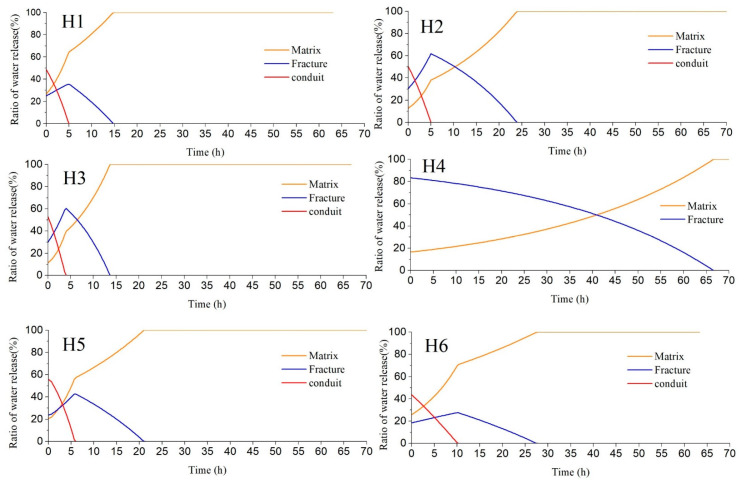
Variation in water release ratio of different media under heavy rain.

**Figure 7 ijerph-18-05775-f007:**
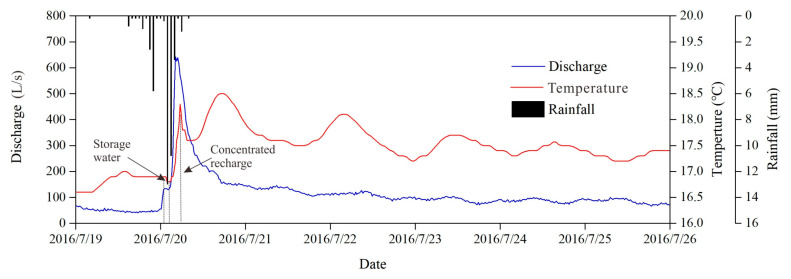
Typical discharge and water temperature variation chart under heavy rain.

**Figure 8 ijerph-18-05775-f008:**
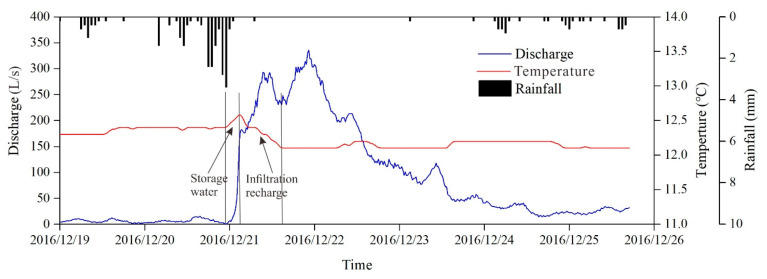
Typical discharge and water temperature variation chart under light rain.

**Figure 9 ijerph-18-05775-f009:**
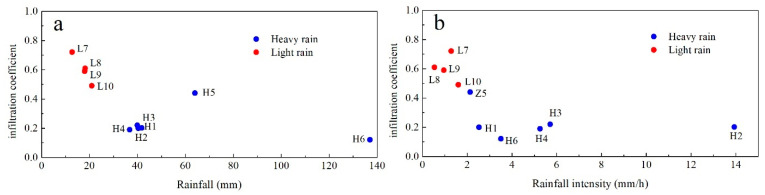
(**a**) Relation between rainfall infiltration coefficient and rainfall amount; (**b**) Relation between rainfall infiltration coefficient and rainfall intensity.

**Figure 10 ijerph-18-05775-f010:**
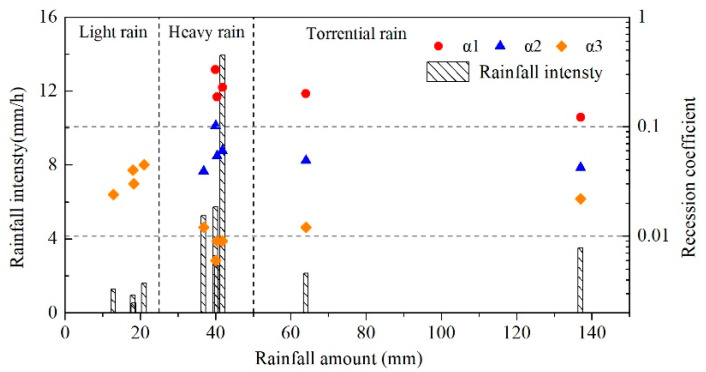
Recession coefficients under different rainfall conditions.

**Figure 11 ijerph-18-05775-f011:**
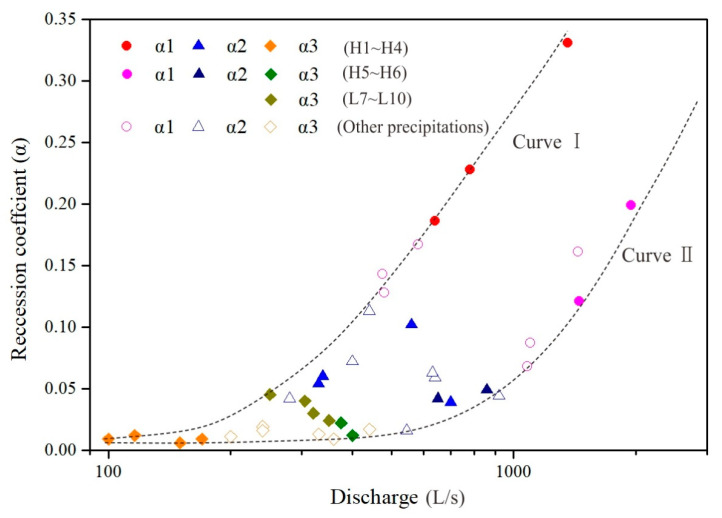
Relationship between recession coefficients and discharge flow rate at the beginning of each recession stage.

**Figure 12 ijerph-18-05775-f012:**
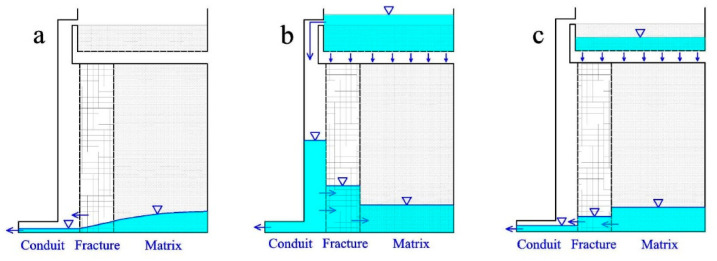
Stages of water charging and releasing process under short-time heavy rainfall condition. (**a**) No rainfall for a long time; (**b**) Recharge in short-time heavy rain; (**c**) Recession after heavy rain.

**Figure 13 ijerph-18-05775-f013:**
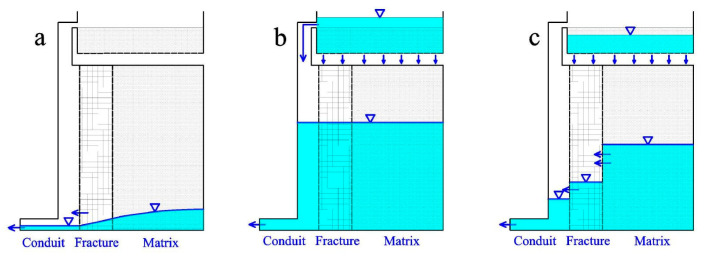
Stages of water charging and releasing process under long-time heavy rainfall condition. (**a**) No rainfall for a long time; (**b**) Recharge in long-time heavy rain; (**c**) Recession after long-time heavy rain.

**Figure 14 ijerph-18-05775-f014:**
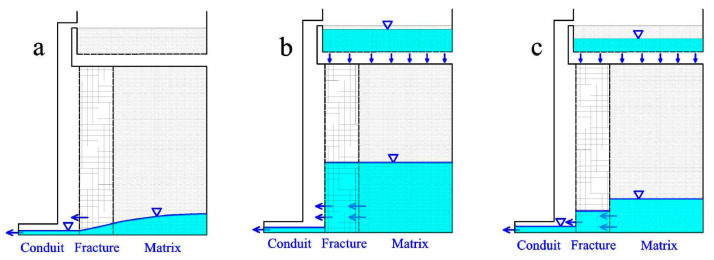
Stages of water charging and releasing process under light rainfall condition. (**a**) No rainfall for a long time; (**b**) Recharge in light heavy rain; (**c**) Recession after light rain.

**Table 1 ijerph-18-05775-t001:** Information of tracer tests in Daiye Cave.

Test Number	Tracer Mass(g)	Injection Point	Recovery Point	Average Discharge (L/s)
1	160	Sinkhole ④	Daiye Cave	60
2	370	Sinkhole ④	Daiye Cave	650

**Table 2 ijerph-18-05775-t002:** Results of tracer tests.

Test Number	Mean Velocity (m/h)	Reynolds Number	Recovery Rate (%)	Conduit Diameter (m)	Cross-Sectional Area (m^2^)	Total Volume Estimate (m^3^)
1	32.72	21,389	35.534	2.683	5.6538	6671.5
2	251.65	165,010	51.664	2.691	5.6878	6711.6

**Table 3 ijerph-18-05775-t003:** Characteristic parameters of rainfall discharge curve.

Rainfall. Number	Total Rainfall (mm)	Initial Discharge (L/s)	Post-Recession Discharge (L/s)	Peak Discharge (L/s)	Response Time (h)	Lag Time (h)	Delay Time (h)	Rise Time (h)	Recession Time (h)
H1	40.4	45	65	642	2	3	166	5	161
H2	41.8	10	40	778	1	2	129	2	127
H3	40	13	80	1420	1.5	5	145	3	142
H4	36.8	100	55	700	2	5	71	4	67
H5	63	85	155	2065	2	2	91.5	4	87.5
H6	137	70	100	1519	1.5	1	72	9	63.5
L7	12.8	50	17	310	4.5	18	130	17	100
L8	18.2	5	35	301	9	19	133	18	115
L9	18	2	14	335	6	15	110	10	88
L10	21	23	14	295	4	29	144	28	100

**Table 4 ijerph-18-05775-t004:** Recession coefficients under different rainfall conditions.

Rainfall Number	Rain Type	First Stage	Second Stage	Third Stage
H1	40.4 mm rainfall in 16 h (20 July 2016)	Q_1_ = 640 × e^−0.186t^	Q_2_ = 330 × e^−0.054t^	Q_3_ = 170 × e^−0.009t^
H2	41.8 mm rainfall in 3 h (7 June 2016)	Q_1_ = 780 × e^−0.228t^	Q_2_ = 338 × e^−0.06t^	Q_3_ = 100 × e^−0.009t^
H3	40 mm rainfall in 7 h (22 April 2018)	Q_1_ = 1360 × e^−0.331t^	Q_2_ = 560 × e^−0.102t^	Q_3_ = 150 × e^−0.006t^
H4	36.8 mm rainfall in 7 h (22 May 2018)	——	Q_2_ = 700 × e^−0.039t^	Q_3_ = 116 × e^−0.012t^
H5	64 mm rainfall in 30 h (12 August 2017)	Q_1_ = 1950 × e^−0.199t^	Q_2_ = 860 × e^−0.049t^	Q_3_ = 400 × e^−0.012t^
H6	137 mm rainfall in 39 h (28 June 2016)	Q_1_ = 1450 × e^−0.121t^	Q_2_ = 650 × e^−0.042t^	Q_3_ = 375 × e^−0.022t^
L7	12.8 mm rainfall in 10 h (29 November 2016)	——	——	Q_3_ = 350 × e^−0.024t^
L8	18.2 mm rainfall in 34 h (12 March 2017)	——	——	Q_3_ = 320 × e^−0.03t^
L9	18 mm rainfall in 19 h (20 December 2016)	——	——	Q_3_ = 305 × e^−0.04t^
L10	21 mm rainfall in 13 h (5 April 2018)	——	——	Q_3_ = 250 × e^−0.045t^

**Table 5 ijerph-18-05775-t005:** Ratio of water release from different media.

Rainfall Number	Conduit			Fracture			Matrix			Total Volume (m³)
	Duration (h)	Volume (m³)	Proportion (%)	Duration (h)	Volume (m³)	Proportion (%)	Duration (h)	Volume (m³)	Proportion (%)	
H1	5	2436	5.2	14.7	3723	8.0	100	40,496	86.8	46,656
H2	5	3390	9.0	24	7848	20.8	120	26,496	70.2	37,735
H3	4	4371	8.8	13.7	7980	14.9	100	40,990	76.3	53,702
H4	——	——	——	66	41,000	62.7	100	24,400	37.3	65,400
H5	5.3	10,450	9.5	21	14,697	13.4	100	84,648	77.1	109,795
H6	10	11,655	15.1	27.7	10,472	13.6	100	54,814	71.3	76,942
L7							100	47,966	100	47,966
L8							100	36,690	100	36,690
L9							100	27,134	100	27,134
L10							100	19,930	100	19,930

## Data Availability

Data available on request due to restrictions e.g., privacy or ethical.

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
