# Peer review of "Responses of Spring Discharge to Different Rainfall Events for Single-Conduit Karst Aquifers in Western Hunan Province, China"

_ijerph, 2021, doi:10.3390/ijerph18115775_

Round 1

Reviewer 1 Report

The article, as revised, appears to be a good research work. Now the more detailed introduction is adequate and the article has generally been improved. The figure 6, related to the "Variation of water relase ratio of different media under heavy rain", is now better explain. My decision is therefore "accept in present form".

Best regards

Author Response

Thank you again for your constructive comments, which played a decisive role in the improvement of our manuscript.

Best regards

Reviewer 2 Report

Dear authors, thank you for your revision of the manuscript, which make it more readable and more extended as scientific soundness, especially regarding the introduction part in which a wider range of hypothesis and literature application has been provided.

Some little suggestions more: Please control the punctuation and orthography and the spaces distribution along the manuscript; for example, at rows 46-47 (but you can find the same in the following parts of the text):

"Many methods are applied to identify karst groundwater system, such as hydrogeo-chemical analysis[9], environmental isotope method[10] and artificial tracer test[11]. "

missing spaces between words and references.

Now the introduction chapter is more detailed and suitable for the publication purposes.

IN the study area chapter please correct some orthographic errors and add the references as indicated in the pdf file.

In the material and method section you added some explanation more, and this is good for the comprehension of the results; in the tracer test description, you should indicate how did you measure the discharge in the stream entering the sinkhole. A question: in my opinion the flow rate was different in the two test depending on the different amount of precipitation in the two events: have you measured the rainfall amount? I suppose the events are reported in the figure 3, but I think that a clear and explicit indication of the total amount of rainfall in the two events would enhance the understanding of the processes involved.

The result chapter is perfectly intelligible. A suggestion for your future research: a time series analysis coupled with statistic methods could make more readily apparent and interpretable the connection between rainfall events, discharge response and tracer test BTC. Maybe in the future you can try to find more correlations among your datasets.

The discussion part  has been almost completely re-written and results more readable, with different issues linked together.

the conclusions are supported by the presented results.

In the attached file you can find other (minor) suggestions and corrections.

Author Response

Thank you for your and reviewers’ comments on our manuscript entitled “Responses of Spring Discharge to Different Rainfall Events for Single-Conduit Karst Aquifers in Western Hunan Province, China” (ID: ijerph-1231591). We have revised the manuscript carefully in addressing all the comments. Our point-to-point responses to the comments were provided below with corresponding revisions provided in the manuscript with the revised portions marked in red.

Responds to the reviewer’s comments:

Reviewer #2

Q1: Some little suggestions more: Please control the punctuation and orthography and the spaces distribution along the manuscript; for example, at rows 46-47 (but you can find the same in the following parts of the text):

"Many methods are applied to identify karst groundwater system, such as hydrogeochemical analysis[9], environmental isotope method[10] and artificial tracer test[11]."

missing spaces between words and references.

R1: All the spaces between the words and references have been added, and relevant references provided by reviewers have been added in lines 47-49.

Q2: In the study area chapter please correct some orthographic errors and add the references as indicated in the pdf file.

R2: We have checked the chapter of the "Study area" and corrected the orthographic mistakes.

The title “materials and methods” was modified as “Materials and Methods” in line 147.

The word “which” was modified as “that” in line 258.

In addition, relevant reference has been added in line 129.

Q3: In the material and method section you added some explanation more, and this is good for the comprehension of the results; in the tracer test description, you should indicate how did you measure the discharge in the stream entering the sinkhole. A question: in my opinion the flow rate was different in the two test depending on the different amount of precipitation in the two events: have you measured the rainfall amount? I suppose the events are reported in the figure 3, but I think that a clear and explicit indication of the total amount of rainfall in the two events would enhance the understanding of the processes involved.

R3: It is a pity that we did not monitor the discharge at the entrance of the sinkhole and the average discharge refers to the discharge of the Daiye cave spring, but rainfall has been monitored during both tracer tests which can reflect the discharge in the stream entering the sinkhole. The first group of tracer test was carried out without rainfall, we added it on line 194. The second group of tracer tests was carried out when the total rainfall was 36.8mm and the tracer was injected during the rainfall(line 195).

Q4: The result chapter is perfectly intelligible. A suggestion for your future research: a time series analysis coupled with statistic methods could make more readily apparent and interpretable the connection between rainfall events, discharge response and tracer test BTC. Maybe in the future you can try to find more correlations among your datasets.

R4: Thanks for your suggestions. In our future studies, time series analysis method and other methods will be applied to explore the relationship between rainfall events, discharge response and tracer test BTC. 

Reviewer 3 Report

Chang et al. submitted the paper entitled "Responses of Spring Discharge to Different Rainfall Events for Single-Conduit Karst Aquifers in Western Hunan Province, China'' to the International Journal of Environmental Research and Public Hearth. The paper investigates the rainfall-discharge relation of a karst characterized by a single condit. The article is supported by tracer experiments and rainfall/discharge events with convincing argumentation. It is unclear why this paper was not submitted to MDPI hydrology as it better suits there. This paper does not address the impact of groundwater on public health as it is mainly a hydrology investigation ... Furthermore, water quality is not addressed either. However, the paper provides sufficient proof of study investigation with interesting measurement and proposed discharge mechanism that could explain the difference with light and heavy rainfall impact. What still missing is a short discussion of the limitation of the investigation method, the additional record.observation need such as the fracture network description, an observation of the karst system such as a geological description, the impact of having only 1 rainfall gauge station observation., the assumption of having only 1 conduit etc … As this paper seems to be re-submission there have already been major corrections in the paper. It presents only minor changes required to be submitted. See further comment for improvement below.

Questions

Q1

Why publishing in International Journal of Environmental Research and Public Health, MDPI hydrology will be more appropriate ... as I don't see the clear connection to public health of your paper. This paper does not connect to public water need ... or water quality ...

Q2

Line 262 "rainfall center to peak discharge". How was that point estimate ? Based on Figure 1 there is only 1 rainfall monitoring station...

Q3

Based on Line 294, you supposed that the fracture water will directly goes to discharge (or the conduit). Is the part matrix --> fracture that direct ? Matrix may feed fracture in the rock, there is no guarantee that fracture exactly connects to the conduit ... It may depend on the geometry of the quartz and there is no 3D map of it. Would be useful to have a description of the fracture kartz ...

Q4

What the potential impact of the rainfall monitoring station (as not on the recharge area) on your results ... There is a probability that rainfall occurs at the station but is not present in the recharge area or being recorded as heavy at the station but not in the recharge area. That could be discussed at end. Furthemore, limitations of the study should be mentioned for example the absence of 3D karst description, fracture location on the maps and the assumption that there is a single conduit in the study area ...

Q5

What could be the impact of the karst depression near the Daiye Cave ..

Q6

What the impact of the probably permeable Cambiran layer ... strong infiltration until the Cambrian layer??. Without further investigation of the karst network, it is difficult to know the vertical path of the water flow ...as illustrated by the cross section Figure 1.

Figures

Figure 1

In your legend, geological formation is noted as strong/Medium and Weak. You should keep consistent and label it Ordovician/Canbrian and remove that labelling from the main map ... Furthemore, it is not clear what corresponds to Weak/Medium and strong karst aquifer ... the stiffness of the rock, the storage capacity ... it too vague term

You need to show your watershed or the elevation that we can understand the surface water circulation path more clearly

What is the pink dashed-point line ?? Red dashed point line mean the recharge area=watershed, is that the regional watershed

Figure 2

Dischrge --> Discharge

The Test 1/2 should be labeled Tracer test, those curve represent the tracer record but when the tracer injection was done (Precise the time/date on your graph)

Figure 4

Re-precise in your caption which Z is considered as heavy rainfall and the one as moderate in ().

--> i.e.

(Z1 ~ Z6 are considered as heavy rainfall and Z7~Z10 as light rainfall events)

Figure 12-13-14

Precise exactly what mean a,b,c ...

Tables

Table 3

Table 3. Characteristic parameters of rainfall discharge curve.

-->

At what location the total rainfall is estimated ? the gauge station ?

Specific comments

Line 106-107

Finally, conceptual models were concluded to explain the hydraulic property

-->

Finally, conceptual models were proposed to explain the hydraulic property

Line 133

karst mesa ??

Line 127

Arrange the Table location that it fit on one page not split

Line 245

Not clear what "develops maturely " means in that context. USe a more precise term to describe your meaning.

Line 332-2337

please revise that this record is done during winter period which explain that the gw temperature is higher than the surface water. Would have been better to have a summer light rain for comparison ...

Author Response

Best regards !

This manuscript is a resubmission of an earlier submission. The following is a list of the peer review reports and author responses from that submission.

Round 1

Reviewer 1 Report

Dear author,

The stating idea of the manuscript sounds very interesting but the manuscript suffers from lots of weakness. Please find some comments in the file enclosed.

Sincerly,

Reviewer 2 Report

Dear authors, please find attached comments and suggestions.

Reviewer 3 Report

The paper deals with an important issue regarding karst hydrogeology, and points out to the response of spring discharge to different types of precipitation events. The use of recession curves to study the aquifer behavior is well known in the scientific literature.

The authors provide challenging issue at the end of the abstract, namely to perform more insight understand on hydraulic behaviors of karst spring under different rainfall events. Not entirely fulfilled, by the way, in the rest of the text.

There are several orthographic errors along the text, please check it more carefully. Moreover, please check CAREFULLY the punctuation, commas and the readability of some sentences of difficult interpretation. Reading certain parts of the manuscript (in particular the discussion) is very very complicated! I suggest, after all, a strong language review that make the reading of the manuscript more fluent.

The introduction is well written, with several references especially on the hydrographs study and the recession limb; a lack of references and a lack of discussion in the final part, where the authors introduce the recharge area evaluation issue. In my opinion, more detailed discussion about the use of tracer tests in karst areas and the use of other techniques to identify the recharge area (i.e. geochemistry and stable isotopes) is needed to improve the scientific soundness of the paper.

Methods are rather well exposed, except for the tracer tests. Which number of tests did you perform? How? Where did you inject the tracer? Which amount? Where did you detect tracer concentration? How lasted the tests? Please provide more and more information about this crucial issue.

In the methods indication of software or model utilized to interpret the tracer tests is missing.

In the results, some parts can be moved in the methods chapter in order to improve the readability.

I suggest to add a discussion about the shape of breakthrough curves in the two periods. Why these shapes? You can read the literature to answer accordingly.

You should discuss whether the aquifer volume obtained by the tracer tests is in line (or not) with the aquifer properties, distance sinkhole/spring and so on. Did you expected this value?

In the results the part of rainfall events and hydrograph analyses is very detailed and extensive; I suggest to enlarge the characters in the figures to improve the readability.

During the discussion you assert that some water infiltrates the conduits and other infiltrates fissures and porous matrix: for me, it is not so clear how you quantify this distinction; only by hydrographic analysis? is it so? perhaps you should base this statement even on other evidence... please clarify it and discuss

Rows 425-442 to increase the readability and comprehension of such a scheme and its description (that is a very important part of the paper), I think it's better indicate that numbers 1-2-3 represent different stages of the precipitation process (I suppose...) and that L and H represent light and heavy rain. Please modify the description in these rows because so it's very difficult to understand. This is a key point of the paper, so I hope you put the right attention to this. From the figure it's not clear what is the difference between situation L1 and H1: can you describe it better? can you use different symbols in the picture?

Conclusion chapter: since the first point of the conclusions regards the tracer test, I am increasingly demanding you again to be more explicit and give more detailed description about tracer tests both in the methods and the results/discussion.

Round 2

Reviewer 1 Report

I read the manuscript for a second time, and I do not find a significative improvement compared with the first version.

The data set seems to be good enough for a publication, but I would recommend to work harder on how to extract interesting information and discussion. All my comments and som suggestions can be finded directly in the modified version of the manuscript.

Reviewer 2 Report

I appreciate the work done by the authors but they still have to make important revisions of the article, in the description of some methodological parts, as described in detail in the annex.

Reviewer 3 Report

Dear authors, the paper has really improved with respect to the first sending of the manuscript. Unfortunately there is an important suggestion regarding the introduction chapter that you have not follow; maybe my comments weren't so clear...

So I ask you to add an important part to the introduction chapter.

In particular, in the revision form I wrote: " In my opinion, more detailed discussion about the use of tracer tests in karst areas and the use of other techniques to identify the recharge area (i.e. geochemistry and stable isotopes) is needed to improve the scientific soundness of the paper. " and you replied: " Response 2: Thank you to the reviewer suggestions for the introduction. At the same time, we added how to identified the recharge area of groundwater system by tracer test. "

I meant that this part is to be inserted in the introduction; I didn't find where you inserted how to identify the recharge area with tracer (maybe in results?), but, in any case, I asked specific mention to other techniques used to identify the recharge area. Please add this part with specific references.